# DisControlFace: Adding Disentangled Control to Diffusion Autoencoder for One-shot Explicit Facial Image Editing

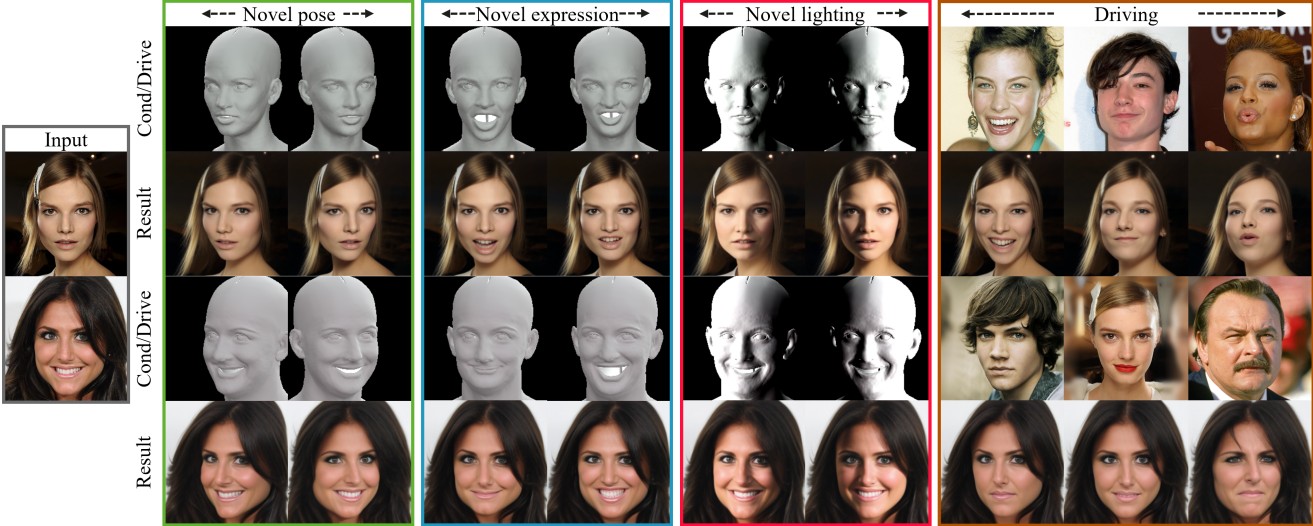

**Figure 1: Our DiscontrolFace can edit the input face image based on explicit parametric control and faithfully preserve the facial semantic appearance under one-shot scenario. Our model can generate realistic and faithful facial image corresponding to diverse pose, expression and lighting conditions and also supports cross-identity face driving.**

## ABSTRACT

In this work, we focus on exploring explicit fine-grained control of generative facial image editing, all while generating faithful facial appearances and consistent semantic details, which however, is quite challenging and has not been extensively explored, especially under an one-shot scenario. We identify the key challenge as the exploration of disentangled conditional control between high-level semantics and explicit parameters (*e.g.,* 3DMM) in the generation process, and accordingly propose a novel diffusion-based editing framework, named DisControlFace. Specifically, we leverage a Diffusion Autoencoder (Diff-AE) as the semantic reconstruction backbone. To enable explicit face editing, we construct an Exp-FaceNet that is compatible with Diff-AE to generate spatial-wise explicit control conditions based on estimated 3DMM parameters. Different from current diffusion-based editing methods that train the whole conditional generative model from scratch, we freeze the pre-trained weights of the Diff-AE to maintain its semantically

deterministic conditioning capability and accordingly propose a random semantic masking (RSM) strategy to effectively achieve an independent training of Exp-FaceNet. This setting endows the model with disentangled face control meanwhile reducing semantic information shift in editing. Our model can be trained using 2D in-the-wild portrait images without requiring 3D or video data and perform robust editing on any new facial image through a simple one-shot fine-tuning. Comprehensive experiments demonstrate that DisControlFace can generate realistic facial images with better editing accuracy and identity preservation over state-of-the-art methods.

## CCS CONCEPTS

• **Computing methodologies → Computer vision problems**.

## KEYWORDS

Facial image editing, Explicit parametric control, Conditional diffusion model

## 1 INTRODUCTION

Facial image editing has long been a hot research topic in the fields of computer vision and computer graphics, where the key challenge is to effectively achieve fine-grained controllable generation of realistic facial images while preserving semantic face priors.

3D Morphable Models (3DMMs) [3, 16] have been widely employed to represent variations in facial shape and texture [4, 16,

*ACM MM, 2024, Melbourne, Australia*
© 2024 Copyright held by the owner/author(s). Publication rights licensed to ACM.
ACM ISBN 978-x-xxxx-xxxx-x/YY/MM
https://doi.org/10.1145/nnnnnnn.nnnnnnn

19, 29, 56–59], whereas their ability to capture personalized facial features is limited and the performance highly depends on the quality and diversity of the 3D face training data. On top of this, subsequent learning-based explicit face models [7, 11, 13, 28, 50, 56–59, 63] achieve controllable generations of dynamic and expressive facial animations by capturing the nuances of facial features under different expressions, poses, and lighting conditions, nevertheless, can neither generate realistic facial appearances that correspond to the animated 3D face geometries nor model refined geometric details in non-facial regions, *e.g.,* hair, eyes, and mouth. Follow-up efforts integrate explicit facial modeling with implicit 3D-aware representations like Neural Radiance Fields (NeRFs) to reconstruct animated realistic head avatars [14, 23, 30, 33, 34, 39–42, 51, 52, 66], which however, heavily rely on 3D consistent data such as monocular portrait videos and tend to exhibit limited generalization.

In contrast, generative face models enable single image reconstruction and editing due to the superior capability in learning rich face priors from in-the-wild portraits. Recent GAN-based approaches [2, 5, 6, 10, 18, 22, 37, 55, 62] incorporate explicit 3D facial priors and implicit neural representations to achieve directed generations of high-resolution, realistic, and view-consistent facial images without the need of 3D face scans or portrait videos. However, those methods mainly provide implicit or limited explicit controls of face generations.

More recently, the diffusion-based framework [21] has emerged as the predominant choice for various generation tasks, owing to its impressive performance and diverse conditioning options. Some approaches have also shown promising enhancements in face reconstruction and various face editing tasks, such as face relighting [43], semantic attributes manipulation [44], and explicit appearance control [12]. Unfortunately, it can be seen that when editing and modifying some specific facial attributes, other facial attributes or editing-irrelevant details often occur unexpected and uncontrollable changes, leading to an incoherent and identity-altered generated face. This prevalent issue can be attributed to that these generative face models struggle to effectively perform disentangled control in the generation process.

In this work, we propose a novel diffusion-based generative framework, namely DisControlFace to achieve one-shot editing of facial images. To generate a photo-realistic, high-fidelity facial appearance corresponding to specific explicit parameters (*e.g.,* 3DMM) while faithfully preserving high-level semantic priors, we particularly focus on enhancing the diffusion model with disentangled conditional control. Specifically, we adopt a Diffusion Autoencoder (Diff-AE) [44] as the generative backbone, which can enable a deterministic image reconstruction by conditioning Denoising Diffusion Implicit Model (DDIM)[53] on the semantic information of the input image. We then specially construct an Exp-FaceNet compatible with the Diff-AE backbone, which further provides multi-scale, spatial-aware DDIM conditioning corresponding to the facial parameters of shape, pose, expression, and lighting. Moreover, we claim that training different DDIM conditoning together, as with existing methods is not conductive to learning disentangled face control. We therefore freeze the pre-trained weights of Diff-AE and accordingly design a random semantic masking (RSM) strategy to enable the training of Exp-FaceNet, by means of which the model can learn explicit parameteric face control independently without

affecting semantically deterministic DDIM conditioning. Also benefiting from this disentangled setting, instead of relying on 3D or video data, we can utilize 2D in-the-wild portrait dataset such as FFHQ [26] to effectively train Exp-FaceNet to learn a robust and generalized capability in explicit face editing. Considering there exists domain gap between the pre-trained face data and the target new face image, which tends to prevent Diff-AE from performing near-exact semantic reconstrcution, we finally introduce an one-shot fine-tuning to Diff-AE so as to restore personal identity and editing-irrelevant details of the input portrait under a subject-agnostic scenario. Our approach not only achieves state-of-the-art (SOTA) qualitative and quantitative results for one-shot explicit facial image editing, but also supports generating realistic and faithful facial appearance of specific individuals in image inpainting, semantic attributes manipulations, and cross-identity face driving (shown in Figure 1).

Our contributions can be summarized as follows:

- We propose a novel diffusion-based generation framework, consisting of a Diffusion Autoencoder (Diff-AE) backbone and an explicit face control network (Exp-FaceNet) for synthesizing photo-realistic, high-fidelity portrait images corresponding to the editing of explicit facial properties only trained with 2D in-the-wild images.
- To the best of our knowledge, we are the first to introduce a weight-frozen pre-trained Diff-AE to explicit face editing pipeline to provide deterministic semantic conditioning, meanwhile designing an effective training strategy to enable the Exp-FaceNet with a disentangled explicit parameteric (*e.g.,* 3DMM) conditioning.
- Our method achieves SOTA generation performance in explicit facial image editing, and also supports various one-shot face editing tasks.

## 2 RELATED WORK

**Generative Face Modeling.** Various GAN-based models [6, 26, 27, 35, 38, 49] have been proposed to synthesize realistic facial images by learning the underlying data priors from large-scale in-the-wild images [6, 38, 49]. However, when it comes to precise control and interpretability, those generative face models fall short compared to explicit parametric models. Given this, several approaches [5, 54, 55, 62] go a step further by integrating explicit parameters and GANs, which can simultaneously generate highly realistic and coherent portraits and achieve fine-grained control of facial appearance. Recently, diffusion models [21, 53] have gained recognition for their superior ability to learn data distributions compared to GANs, and thus have been widely adopted to generate realistic and diverse images in various generation tasks, including facial image synthesis [44, 44]. DiffusionRig [12], a closely related method, introduces pixel-aligned physical properties rendered from explicit parameters estimated by DECA [13] to denoising diffusion process to generate photo-realistic facial images corresponding to target pose, expression, and lighting conditions, which however, highly relies on a personalized fine-tuning (around 20 images) to preserve the facial appearance priors of a specific person . Similar problems widely exist in most conditional diffusion face models, as can be observed where identity shifts and unexpected attributes

alterations may occur during face reconstruction and editing. This can be attributed to the lack of disentangled control capabilities when conditioning diffusion models with both facial semantics and physical information. DisControlFace overcomes this challenge by leveraging a weight-frozen pre-trained Diff-AE to provide semantically deterministic DDIM conditioning and independently training a separate Exp-FaceNet to learn disentangled face control based on 3DMM parameters.

**Conditional Diffusion Model.**    Conditional DDIM enables the Denoising Diffusion Probabilistic Model (DDPM) [21] to generate content consistent with specific control signals though various conditioning manners. Most existing models encode various control information into global conditional vectors, which can be text embedding [45] or semantic embedding [43, 44]. To achieve spatial-aware and precise control of the generation, some approaches (*e.g.,* DiffusionRig [12] and SR3 [48] ) concatenate various spatial conditions and denoised images together as the input of the U-Net noise predictor in each denoising step. However, this form of conditioning requires the U-Net to have a unique input layer, resulting in the model having limited generalization and making it hard to reuse existing well-trained diffusion models. Besides, some other methods specially construct spatial conditioning branches to extract spatial-aligned conditional features and insert them into the U-Net [43, 61, 65]. ControlNet [65] has been widely employed to add various spatial visual guidance (*e.g.,* edge maps, pose maps, depth maps, *etc.*) to text-to-image generation models such as Stable Diffusion [45]. Whereas, it may not suitable for deterministic reconstruction or editing tasks since there still exists uncertainty and randomness in the generation controlled by visual guidance and text prompts. In contrast, our Exp-FaceNet is specially designed to be compatible with a semantic reconstruction DDIM, Diff-AE [44], aiming to better address facial image editing based on explicit control information.

**Learning Specific Facial Priors.**    Effectively extracting the facial appearance priors of the specific person and injecting this global prior into the generation process is crucial for preserving facial semantics such as identity, accessories, hairstyle, and background information in facial image editing. Most existing generative methods address this issue by fine-tuning the network in various settings [15, 24, 36, 47], or designing special optimization strategies, such as identity penalty, face recognition loss, and latent representation editing [1, 31, 60, 64]. In contrast to previous work like [12, 36] which collect personal albums to learn personalized facial priors, in this work, we focus on the subject-agnostic editing scenario, which is more challenging but practical. On the basis of the inherent and maintained facial semantics capturing capability of Diff-AE, only a fast and yet simple fine-tuning using an out-of-domain new face image is needed to faithfully restore the semantic appearance details of the target image during editing.

## 3  PRELIMINARIES

### 3.1   3D Morphable Face Models

FLAME [29] , a popular 3DMM model, can be expressed as $M(\boldsymbol{\beta}, \boldsymbol{\theta}, \boldsymbol{\psi})$ : $\mathbb{R}^{|\boldsymbol{\beta}| \times |\boldsymbol{\theta}| \times |\boldsymbol{\psi}|} \rightarrow \mathbb{R}^{3N}$, which takes shape $\boldsymbol{\beta}$, pose $\boldsymbol{\theta}$, and expression $\boldsymbol{\psi}$ as inputs and outputs a face mesh with $N$ vertices. On this basis, some off-the-shelf 3D face estimators, such as DECA [13]

and EMOCA [7], achieve 3D face reconstruction by regressing individual-specific FLAME parameters from in-the-wild images. We utilize EMOCA to obtain a face mesh corresponding to the input image with enhanced expression consistency, and render it to pixel-aligned explicit conditions. Specifically, given a single image $I$, the coarse branch of EMOCA estimate its corresponding $\boldsymbol{\beta}$, $\boldsymbol{\theta}$, $\boldsymbol{\psi}$, albedo $\boldsymbol{\alpha}$, spherical harmonic (SH) illumination coefficient $\boldsymbol{l}$, and camera $c$, which can be expressed as $E_c(I) \rightarrow (\boldsymbol{\beta}, \boldsymbol{\theta}, \boldsymbol{\psi}, \boldsymbol{\alpha}, \boldsymbol{l}, \boldsymbol{c})$. The detailed branch further outputs a detail vector $\delta$ and computes the displacement map $D$ in UV space, which is specifically expressed as $E_d(I) \rightarrow \boldsymbol{\delta}$ and $F_d(\boldsymbol{\delta}, \boldsymbol{\psi}, \boldsymbol{\theta}_{jaw}) \rightarrow D$.

### 3.2   Diffusion Autoencoders (Diff-AE)

Diff-AE [44] reformulates the traditional diffusion generation model into an autoencoder and captures high-level image semantics for DDIM conditioning, therefore supporting near-exact reconstruction and attribute manipulation of the input image. Specifically, Diff-AE uses a semantic encoder to generate a 512-dimentional latent code $z$ with global semantics of the input image $x_0$. Then, $z$ can be introduced to the reverse deterministic generative process of DDIM to obtain a noisy map $x_T$ which captures the stochastic variations of $x_0$. Last, a conditional DDIM model decodes $(z, x_T)$ to achieve a deterministic reconstruction of $x_0$. By linearly modifying the semantic latent code $z$, Diff-AE can manipulate diverse global semantic attributes, *e.g.,* age, gender, and hairstyle. We introduce Diff-AE to our DisControlFace as the reconstruction backbone and freeze its pre-trained weights to provide semantically deterministic DDIM conditioning during the training of explicit face control.

## 4  METHOD

In the pursuit of a robust one-shot explicit facial image editing, we propose a generative framework, namely DisControlFace, providing DDIM conditioning on disentangled face control between high-level semantics and explicit 3DMM parameters (shown in Figure 2). Specifically, we adopt a weight-frozen Diff-AE as a semantic reconstruction backbone and construct an Exp-FaceNet to provide explicit parametric face control (Sec. 4.1). Furthermore, we design a training strategy to effectively enable the training of Exp-FaceNet (Sec. 4.2). Finally, we adopt an one-shot fine-tuning to improve the semantic consistency and faithfulness of the generated facial image under the subject-agnostic editing scenario (Sec. 4.3).

### 4.1   Exp-FaceNet

On the basis of the adopted Diff-AE reconstruction backbone, an intuitive idea for learning explicit face editing capability is to further build additional DDIM conditioning on 3DMM parameters. Compared to directly adopting the 3DMM parameters as non-spatial control conditions, generating pixel-aligned conditional maps based on those parameters is more conducive and compatible to convolution-based visual representation, which also helps enable fine-grained spatial control for the denoising diffusion process. Given this, we specially construct Exp-FaceNet, an explicit control network compatible with the adopted Diff-AE reconstruction backbone to perform a disentangled spatial conditioning for DDIM-based generation. Here we firs estimate 3DMM parameters from facial images

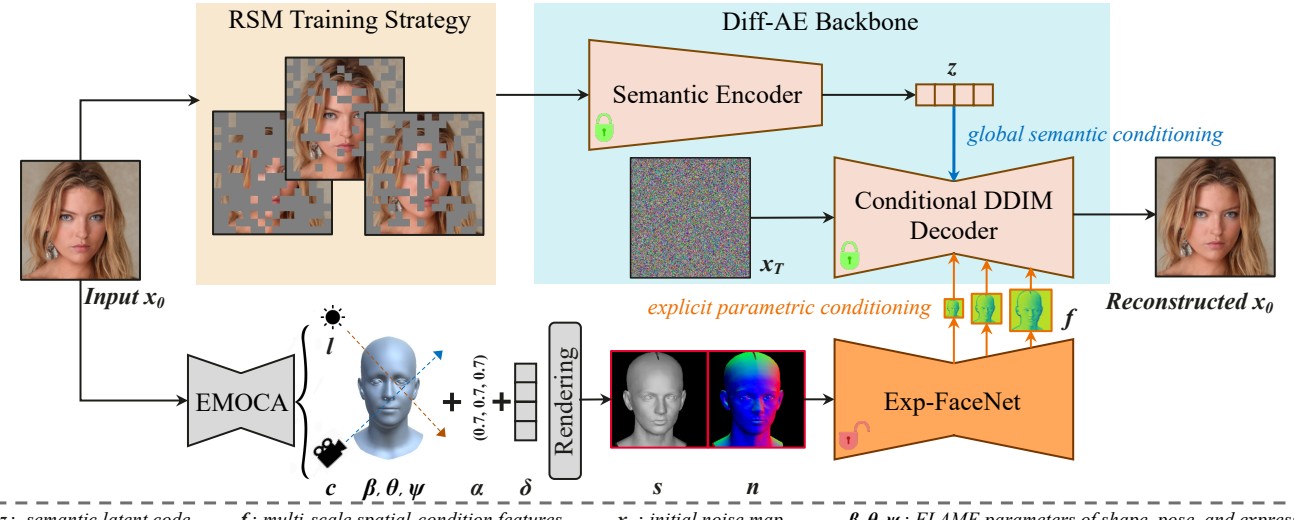

**Figure 2: Pipeline overview.** Our DisControlNet leverages Diffusion Autoencoder (Diff-AE) as the reconstruction backbone freeze its pre-trained weights to maintain the semantic deterministic conditioning capability, which is effective in reducing semantic information shift during the editing of the input portrait image. Then, an explicit face control network, Exp-FaceNet compatible with the Diff-AE is constructed, which takes pixel-aligned snapshots rendered from estimated explicit parameters as inputs and generates multi-scale control features to condition the DDIM decoder. Moreover, a random semantic masking (RSM) training strategy is accordingly designed to enable a disentangled explicit face control of Exp-FaceNet.

and transfer them to the corresponding visual guidance map. Specifically, we use EMOCA [7] to predict FLAME parameters (including shape $\beta$, pose $\theta$, and expression $\psi$), SH illumination parameter $l$, and camera parameter $c$ from an input portrait. Different from those previous work [12, 17, 43], here we also adopt the person-specific detail vector $\delta$ estimated by EMOCA, which can be combined with $\theta$ and $\psi$ to generate the expression-dependent displacement map for refining the face geometry with animatable wrinkle details. To avoid undesired disturbance to facial appearance priors caused by inaccurate and unrealistic appearance estimation, we set the albedo map $\alpha$ to a constant gray value, thereby focusing on controlling the edit of shape, pose, expression, and lighting. We render these explicit parameters into a surface normal snapshot $n$:

$$n = \mathcal{R}(\mathcal{G}(\mathcal{M}(\beta, \theta, \psi)), c, \delta) \tag{1}$$

where the FLAME model $\mathcal{M}$ is used to calculate the 3D face mesh, $\mathcal{G}$ and $\mathcal{R}$ indicate normal calculation function and the Lambertian reflectance renderer, respectively. By means of this, $n$ can reflect the fine-grained geometry of the input face and is compatible with pose parameter $\theta$ and expression parameter $\psi$. Furthermore, we also generate a shading shape snapshot $s$ to illustrate the lighting conditions associated with the SH illumination parameter $l$:

$$s = \mathcal{R}(\mathcal{M}(\beta, \theta, \psi), \alpha, l, c, \delta) \tag{2}$$

Considering U-Net [46] excels at extracting spatial features from images for various vision tasks, we construct Exp-FaceNet in a similar U-shape structure, which takes channel-concatenated snapshots $n$ and $s$ as the input visual guidance map and generates multi-scale deep features for spatial-aware conditioning. Then we feed back the

spatial-condition features outputted by each stage of the U-Net's decoder back to Diff-AE to provide multi-scale conditional control. Please see the detailed architecture in the supplement.

## 4.2 RSM Training Strategy

DisControlFace can be regarded as a DDIM model that is simultaneously conditioned by a global semantic code and multi-scale explicit-control feature maps (see Figure 2). As mentioned before, we freeze the pre-trained weights of Diff-AE to preserve global semantics and enable Exp-FaceNet to learn explicit face control in a disentangled way. However, since Diff-AE backbone already allows a deterministic image reconstruction under this setting, only limited gradients can be generated during error back-propagation, which are far from sufficient to effectively train Exp-FaceNet. Consequently, it is infeasible to train Exp-FaceNet in a traditional conditional DDIM generation form. To address this issue, we design a random semantic masking (RSM) strategy, not for the purpose of representation learning like Masked Autoencoders (MAE) [20], but rather to achieve the training of Exp-FaceNet effectively. Concretely, we divide the input image $x_0$ into regular non-overlapping patches and randomly mask different portions of patches to obtain a masked image $x_0^m$ at different timesteps. By means of this, the semantic latent code $z^m$ encoded by the semantic encoder $\mathcal{E}_\eta$ of Diff-AE only contains fragmented and incomplete content and spatial information of $x_0$. Meanwhile, the spatial-condition features $f$ generated by Exp-FaceNet $\mathcal{F}_\phi$ comprise the fine-grained face shape as well as the camera parameter and lighting condition of $x_0$, which can help to restore the masked face regions in $x_0^m$ in each random denoising timestep. The overall training objective can thereby be

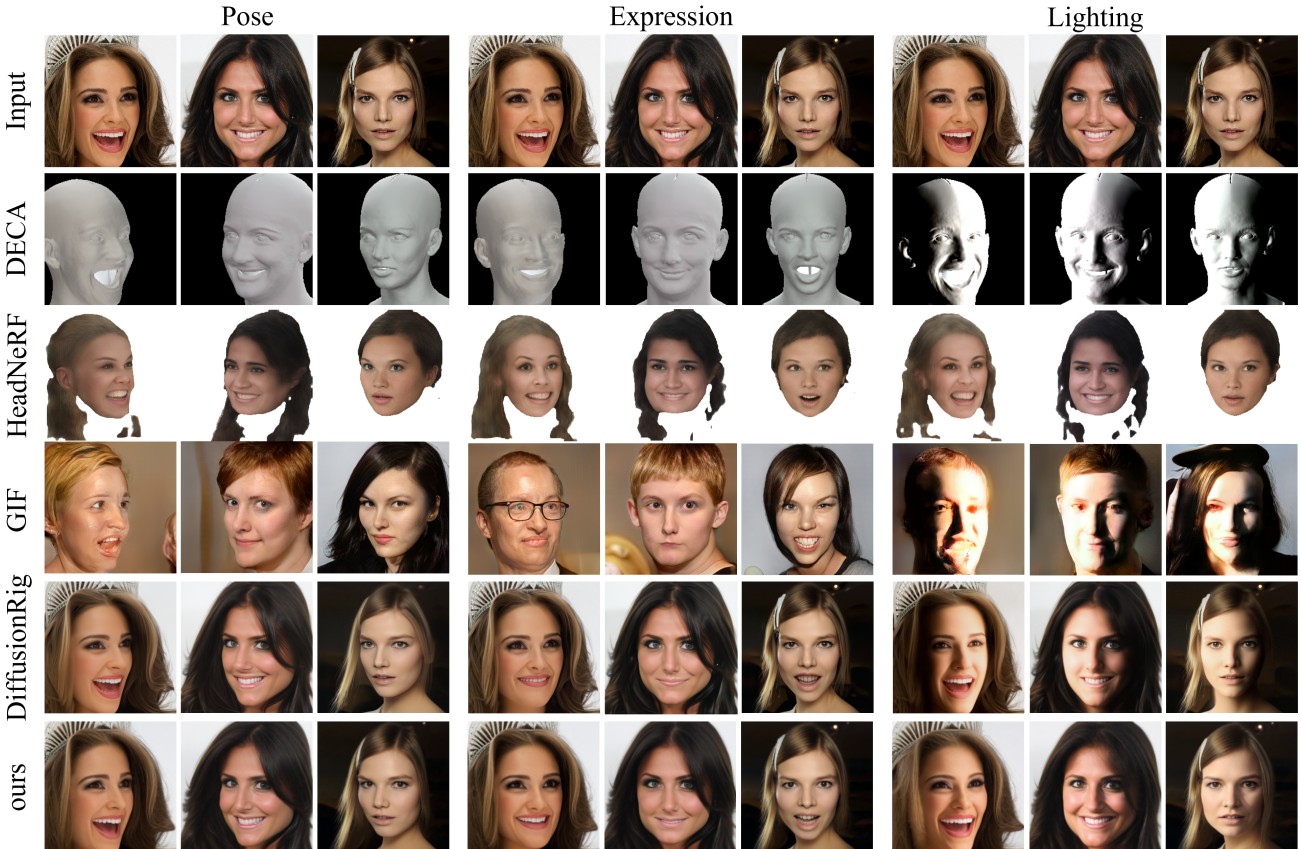

**Figure 3: Qualitative comparison against baselines in one-shot editing. For each selected image, we use EMOCA [7] to estimate the corresponding explicit parameters, then synthesize the edited images using different methods based on the modified parameters of pose, expression, and lighting. We additionally provide the rendered shading shapes in the second row as the references of explicit control conditions. As can be seen, our DisControlFace can edit images that match well with the target control conditions while faithfully synthesizing facial appearances and editing-irrelevant details.**

parameterized as:

$$\mathcal{L} = \mathbb{E}_{x_0, x_0^m, t, \epsilon} [\|\epsilon - \epsilon_\theta(x_t, t, z^m, f)\|_2^2] \quad (3)$$

where $\epsilon_\theta$ is the U-Net of Diff-AE which predicts the noise $\epsilon \sim \mathcal{N}(0, \mathbf{I})$ added in noisy image $x_t$. Throughout the generalized training, we only train $\mathcal{F}_\phi$ and freeze $\epsilon_\theta$ and $\mathcal{E}_\eta$ with the pre-trained weights.

### 4.3 Exploiting One-shot Semantic Priors

At this point, the well-trained Exp-FaceNet is able to explicitly change pose, expression, and lighting of a facial image. However, there still exists identity shift or background changes during the face editing, which is especially evident when the target face for editing lies outside the domain of the pre-trained Diff-AE. Given this, it is meaningful to fully exploit the semantic priors of the to-be-edit image and inject them into the editing process. Concretely, in this stage, we freeze Exp-FaceNet and only fine-tune Diff-AE with the input portrait image using the aforementioned RSM training

strategy. As a result, this one-shot fine-tuning enables the model to faithfully restore the personalized appearance details as well as editing-irrelevant factors such as background and accessories when performing explicit face editing.

### 4.4 Inference Editing

In practice, we first use EMOCA to predict all explicit parameters (mentioned in Sec. 4.1) of the input portrait $x_0$, then we modify the pose parameters $\theta$, expression parameter $\psi$, and SH light parameter $l$ by manually setting target values or directly transferring these parameters from a driving portrait. After this, we calculate the rendered shading shape snapshot $s$ and surface normal snapshot $n$ based on the modified parameters, and further generate explicit control conditions using Exp-FaceNet. On the other hand, since we should keep a consistent generative mechanisms in training and inference, here we also utilized masked input images to provide high-level semantic conditioning for DDIM decoder and accordingly design an intuitive patch masking strategy for inference editing.

|  | ID ↑ | Shape↓ | Pose ↓ | Exp ↓ | Light ↓ |
|---|---|---|---|---|---|
| GIF [17] | 0.22 | 3.0 | 5.6 | 5.0 | 0.40 |
| DiffusionRig [12] | 0.24 | 4.3 | 4.2 | 2.8 | 0.36 |
| Ours | 0.31 | 2.8 | 4.5 | 2.9 | 0.31 |

Table 1: Quantitative comparisons against compared baselines using identity consistency (ID) and DECA re-inference errors on shape, pose, expression, and lighting.

Concretely, for each timestep $t$, we generate the masked image $x_t^m$ by randomly masking the patches of the input image $x_0$ with a linear ratio $\rho_t = 0.75 - 0.5(T - t)/T$, where the number of the inference denoising steps $T$ is set to 20 in this work. Under this setting, we can generate $x_t^m$ with high masking ratios $\rho_t$ to emphasize the facial control of the intermediate denoising result $z_{t-1}$ in the early stage of the inference, and then gradually decrease $\rho_t$ to recover semantic information.

## 5 EXPERIMENTS

### 5.1 Implementation Details

We train the proposed Exp-FaceNet on the FFHQ dataset [26], which consists of 70k in-the-wild facial images. For evaluations, we select the images of the CelebA-HQ dataset [25] to perform one-shot fine-tuning and face editing. To balance generation quality and computational cost, we resize the images to a resolution of $256 \times 256$ for both training and inference. Accordingly, we utilize Diff-AE[1] pre-trained on FFHQ-256 for all experiments. We train Exp-FaceNet for 437,500 iterations, with a learning rate of $1e^{-4}$ and a batch size of 32, while during the one-shot fine-tuning stage, we only fine-tune the pre-trained Diff-AE for 1,500 iterations, with a learning rate of $1e^{-5}$ and a batch size of 4. In all training stages, we use AdamW [32] as the optimizer and set the denoising timesteps to 1,000.

### 5.2 Comparison

**Baselines.** We compare our methods against three generative methods for parametric face image synthesis and editing: HeadNeRF [23], GIF [17], and DiffusionRig [12]. Among these, HeadNeRF is a NeRF-based head model, while GIF and DiffusionRig are both generative face models built upon GAN and diffusion model, respectively. For a fair comparison, we utilize the models pre-trained on FFHQ at a resolution of $256^2$ for each method.

**Qualitative comparison.** We evaluate our DisControlFace against baselines using images from CelebA-HQ and we perform one-shot fine-tuning for all methods. The qualitative comparison results are provided in Figure 3. For all methods, we visualize the editing results of pose, expression, and lighting on three identities. In order to intuitively measure the editing performance, we further give the rendered snapshot $s$ of the shading shape corresponding to the modified explicit parameters as the references of explicit control conditions. (second row of Figure 3). We can find that compared to the other methods, our method synthesizes images with overall best identity consistency and parametric editing accuracy. Specifically, HeadNeRF cannot generate realistic facial appearance as well

---

[1]https://github.com/phizaz/diffae.

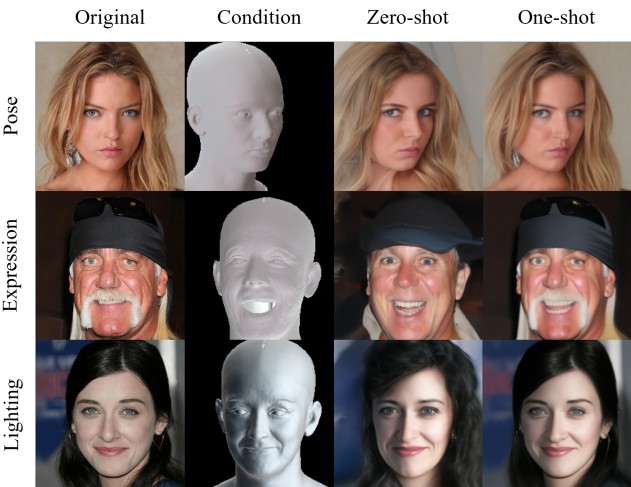

Figure 4: Ablation study on one-shot fine-tuning. DisControlFace can perform zero-shot explicit editing, where however, identity shift still exists. On this basis, adopting a simple one-shot fine-tuning can significantly improve the preservation of face identity as well as other editing-irrelevant semantic information.

as the background of the original image. GIF has a good control ability of parametric face editing, however, is completely unable to preserve the face identity. DiffusionRig, another diffusion-based method achieves better editing results than HeadNeRF and GIF, especially in identity preservation, which can also be attributed to using one-shot fine-tuning to boost the diffusion model with more personal facial priors. Nevertheless, since DiffusionRig trains the whole model together from scratch which prevents the model from learning robust disentangled control, the edited results still have visible identity shift (*e.g.,* eyes, skin color, and hair) especially in lighting editing.

**Quantitative comparisons.** Table 1 (top part) provides the quantitative editing results of all methods on 1000 in-domain images of FFHQ. Following previous work [9, 12, 17], we apply DECA re-inference on edited images and calculate the Root Mean Square Error (RMSE) between the input and re-inferred face vertices as well as spherical harmonics to evaluate the editing accuracy on shape, pose, expression, and lighting. We additionally measure the identity consistency (ID) score by computing the cosine similarity between the deep features generated by ArcFace [8] of the original and edited images. The results indicate that our method achieve the overall best performance on all metrics. Note that DisControlFace outperforms other methods by a large margin in ID score, which demonstrates the superiority of our method in identity preservation during one-shot editing.

### 5.3 Ablation Study

**One-shot fine-tuning.** Based on the deterministic semantic reconstruction capability of Diff-AE [44], DisControlFace can extract global semantics of the input face image. However, since the target

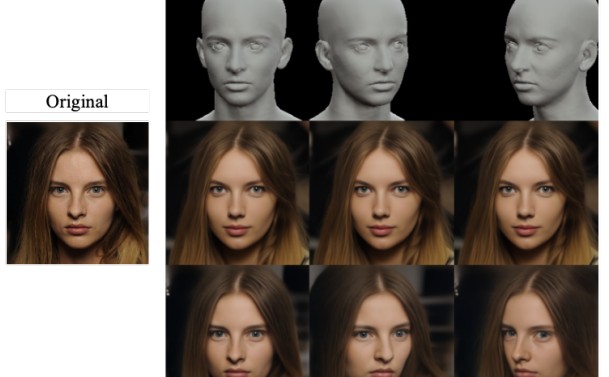

Figure 5: The necessity of the proposed random semantic masking (RSM) training. Without RSM training, it is infeasible to train Exp-FaceNet with explicit face control.

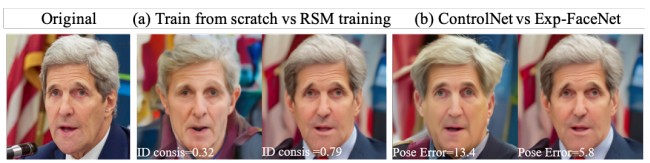

Figure 6: The ablation studies on our disentangled pipeline (a) and Exp-FaceNet structure (b). The disentangled control setting in DisControlFace trained with RSM strategy can significantly improve the identity preservation in explicit editing. Compared to adopting ControlNet with a light-weight decoder and zero convolutions, using our Exp-FaceNet can improve the explicit control accuracy by a large margin.

portrait image tends to have different face priors with pre-trained Diff-AE, there still exists semantic information shift under the zero-shot editing scenario, as shown in Figure 4. Given this, we adopt a simple one-shot fine-tuning on the pre-trained Diff-AE to fully exploit the semantic priors of the to-be-edit image and inject them into the editing process. We can observe that by means of this, the face identity and other high-level semantic information (*e.g.*, background, accessories, and hair) can be well preserved.

**Effectiveness of RSM training.** In this paper, we achieve a disentangled training of the proposed Exp-FaceNet by freezing the pre-trained weights of Diff-AE and according designing a RSM training strategy. To demonstrate the necessity of the proposed RSM training, we separately train Exp-FaceNet with and without random patch masking for the input image of semantic encoder of the Diff-AE backbone. Figure 5 shows that both training strategies enable the model to reconstruct the input image. However, only the model trained with PSM strategy can generate images with novel poses. This result is consistent with our claim that since the pre-trained Diff-AE backbone can already allow deterministic image reconstruction, limited gradients can be generated during error

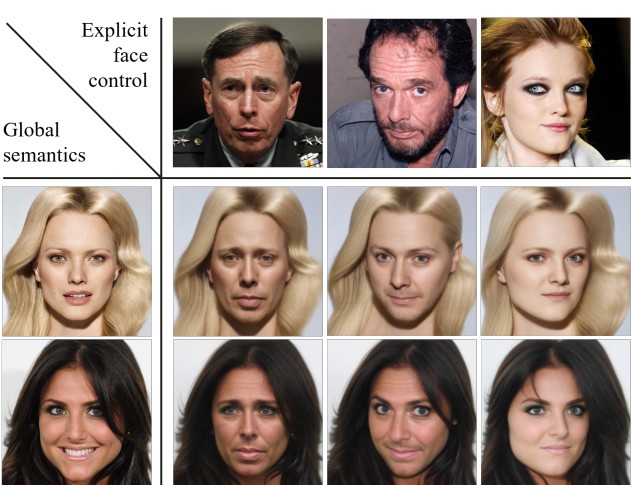

Figure 7: Visualization of disentangled face control. We separately utilize the encoded global semantic code of one image and the estimated 3DMM parameters of another image to provide semantic conditioning and explicit parametric conditioning in facial image generation.

back-propagation for an effective training of Exp-FaceNet.

**Effectiveness of our disentangled pipeline** To further demonstrate the advantages of our method over traditional pipeline in terms of disentangled control, we train the whole model (Diff-AE+Exp-FaceNet) from scratch without using RSM training. The result in Fig. 6 (a) shows that our disentangled pipeline can significantly reduce identity shift, both visually and in terms of quantitative metrics.

**Exp-FaceNet structure.** Our Exp-FaceNet is inspired from ControlNet [65], a popular deep network which has been widely employed to add various spatial visual guidance (*e.g.,* edge maps, pose maps, depth maps, *etc.*) to Stable Diffusion (SD) [45] for text-to-image generation. However, there exists many differences between two models. First, ControlNet is specially designed for SD which can generate specific content based on the input prompts, while there still exists uncertainty and randomness in the generation controlled by visual guidance and text prompts. In contrast to this, this paper focuses on semantics preservation and explicit editing of the input portrait image. Given this, our Exp-FaceNet is designed to be compatible with a Diff-AE backbone, aiming to provide semantically deterministic DDIM conditioning. Second, we construct Exp-FaceNet as a U-Net instead of adopting zero convolutions introduced in ControlNet. With this setting, we can endow Exp-FaceNet with strong capability of U-shape models in extracting spatial deep features from pixel-aligned visual guidance map (rendered explicit snapshots in this work). Here we compare the generation performance between using Exp-FaceNet and ControlNet to learn explicit face control. The results in Fig. 6 (b) show that compared to ControlNet, our Exp-FaceNet can help to edit the face image with less DECA re-inference error and visualized better identity consistency.

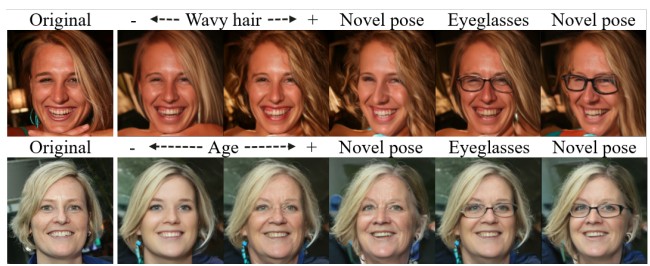

**Figure 8: Synthesis results in semantic manipulation. We follow Diff-AE [44] to manipulate the global facial attributes (age, hairstyle, and eyeglasses) by linearly editing $z$. Owing to the disentangled control mechanism, our model can simultaneously perform semantic manipulation and explicit editing of the input facial image.**

## 5.4 Visualization of Disentangled Control.

To further analyze the disentangled conditioning of our DisControl-Face, we show the synthesis results of mixed conditioning generation in Figure 7. Specifically, we replace the encoded global semantic code of facial image $A$ with global code extracted from another image $B$, while fixing all the estimated all explicit face parameters of $A$. We can observe that the synthesis face has the same face shape, pose, expression, and lighting condition with image $A$. Meanwhile, all facial appearance priors like skin color, eyes, and lips color as well as non-facial high-level semantics such as background and hair of image $B$ have been successfully transferred to the synthesized image. Note that we can achieve robust facial semantics transfer without requiring a few-shot fine-tuning on the personalized images of an specific individual as adopted by DiffusionRig [12]. This result can intuitively show the effectiveness of the disentangled face control capability of DisControlFace. Also benefiting from this, the proposed DisControlFace can preserve original high-level semantics while performing fine-grained explicit face editing.

## 5.5 Applications

**Semantic manipulation.** Since we adopt Diff-AE [44] as the reconstruction backbone and freeze the pre-trained weights without tuning, our DisControlFace inherits the encoding capability of global facial semantics. Therefore, compared to previous diffusion-based model such as DiffusionRig [12], our model also supports manipulating face attributes (*e.g.,* hairstyle, age, and accessories) of the input portrait by linearly editing global semantic codes. The visualized results of both semantic manipulation and explicit editing are shown in Figure 8 , which once again demonstrate the effectiveness and flexibility of the proposed disentangled conditional generation mechanism.

**Image inpainting.** Benefiting from the proposed RSM training strategy, our model inherently supports image inpainting. Figure 9 shows the results of zero-shot inpainting on center-masked facial images as well as the subsequent editing. We can observed that the restored face in the inpainted image is smooth and natural, also has good similarity with the original face. Besides, the explicitly edited image still shows a consistent identity with the restored

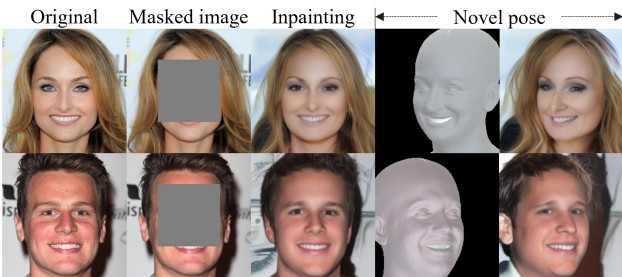

**Figure 9: Zero-shot inpainting and subsequent explicit editing on images from CelebA-HQ. DisControlFace can restore the masked face regions smoothly and naturally. On this basis, Our method can further edit the restored facial images based on the modified explicit 3DMM parameters.**

facial image. Meanwhile, since it is inappropriate to use masked images to perform one-shot fine-tuning on Diff-AE, some editing-irrelevant high-level semantics such as hairstyle and background might not be well preserved in the generated images.

## 6 LIMITATIONS AND CONCLUSION

**Limitations and future work.** In this work, both adopted Diff-AE and constructed Exp-FaceNet were trained using the FFHQ dataset, which consisting 70,000 in-the-wild images. However, this data size is still not sufficient to train the model to learn a more generalized face priors. It can be expected that collecting much more face data with abundant face conditions (*e.g.,* pose and expression) for training can substantially improve the editing performance, especially for some challenging tasks like zero-shot explicit editing. We adopt EMOCA to estimate explicit face parameters of the input image and generate spatial-aware conditions representing control information based on them. However, EMOCA struggles to model the detailed geometry of eyeballs as well as some extreme expressions, which could hinder the model from restoring the corresponding facial details when generating edited images. Furthermore, our editing framework is constructed as a denoising diffusion pipeline, which therefore is still unable to compete with GANs in terms of generation speed. In the future, with the development of fast sampling algorithms, we expect the generation time of our model to further decrease.

**Conclusion.** We have presented DisControlFace, a novel diffusion framework for one-shot facial image editing. Through exploiting disentangled conditioning on high-level semantics and explicit 3DMM parameters in the generation process, our model excels in explicit and fine-grained face control while preserving semantic information and facial priors in face editing. This may boost a series of related applications including various semantic and explicit facial image editing, zero-shot image inpainting, and cross-identity face driving.

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
