# OpenReview forum: "DisControlFace: Adding Disentangled Control to Diffusion Autoencoder for One-shot Explicit Facial Image Editing"
_acmmm.org/ACMMM/2024/Conference — MM2024 Poster_

### Official Review · Reviewer_odPW · 2024-05-23

**Rating:** 4
**Confidence:** 3

**Summary:**

This paper proposes DisControlFace which supports fine-grained control of generative facial image editing under a one-shot scenario.
DisControlFace is built on the top of a Diff-AE. Based on it, this paper introduces Exp-FaceNet with a random semantic masking training strategy to learn to disentangle face control.

**Strengths:**

- This paper is well-written and is easy to follow.
- This paper suggests freezing Diff-AE and implementing a random semantic masking training strategy to enable training different DDIM conditioning together. The effectiveness of this approach is verified through ablation studies, highlighting its potential to inspire future research.
- DisControlFace achieves sota qualitative and quantitative results for one-shot explicit facial image editing.

**Limitations:**

- The improvement in lighting change in Fig. 3 is remarkable when compared with DiffusionRig. Some instructive discussion of this is missing, e.g., does it benefit from the RSM training strategy and how? On the contrary, the improvement is not obvious in pose and expression. It is recommended to highlight the improvment and add more comparisons to support the claim.

- I appreciate the visualization of disentangled face control in Fig.7. However, It seems that the ID is not controlled by global semantics and the ID information will be restored by the one-shot fine-tuning. An ablation study of disentangled face control w.o one-shot fine-tuning is recommended for a better understanding of the ID control in DisControlFace.

Typos:
Line 750. PSM -> RSM;
Line 835. all the estimated all explicit … -> all the estimated explicit

**Suitability:**

3

---

### Official Review · Reviewer_Shn4 · 2024-05-24

**Rating:** 1
**Confidence:** 4

**Summary:**

This paper focuses on fine-grained control in facial image editing while maintaining realistic facial appearances and consistent semantic details. The key contributions include a diffusion-based editing framework, DisControlFace, leveraging a Diffusion Autoencoder (Diff-AE) as the backbone for semantic reconstruction and an Exp-FaceNet for spatial-wise explicit control conditions based on 3DMM parameters. Experimental results show that the proposed method outperforms existing methods in controllable facial image editing.

**Strengths:**

1. **Nice figures**: The pipeline overview and experimental results are very polished.
2. **Rich paper review**: The related work review is rich and diverse.

**Limitations:**

1. **Missing details and mistakes**:

   - > To address this issue, we design a random semantic masking (RSM) strategy, not for the purpose of representation learning like Masked Autoencoders (MAE) [20], but rather to achieve the training of Exp-FaceNet effectively.

     Sections 4.2 and 4.3 are unclear and lack sufficient detail. What are the main differences between RSM and MAE?

   - [Figure 3] "DECA"->"EMOCA"
   - [Line 677] Which is the "top part"? Any other bottom part?

2. **Typos**: There are several grammar mistakes and typos.

   - [Line 121, 137,140] "generations"->"generation"
   - [Line 168] "conditoning"->"conditioning"
   - [Line 173, 202] "parameteric"->"parametric"
   - [Line 182] "reconstrcution"->"reconstruction"
   - [Line 243] "though"->"through"
   - [Line 309] "dimentional"->"dimensional"
   - [Line 347] "firs"->"first"

**Suitability:**

3

---

### Official Review · Reviewer_nN7V · 2024-05-24

**Rating:** 3
**Confidence:** 3

**Summary:**

The paper introduces a diffusion-based generative framework for explicit fine-grained control of generative facial image editing. This framework utilizes a pre-trained Diff-AE model with an Exp-FaceNet to condition the diffusion process based on 3DMM parameters. The authors also introduce a Random Semantic Masking (RSM) training strategy to enable a disentangled explicit face control of the Exp-FaceNet. The proposed method achieves explicit facial image editing and also supports various one-shot face editing tasks. The comparison experiments show the effectiveness of the proposed method over existing methods.

**Strengths:**

1. The random semantic masking training strategy effectively enables disentangled editing based on 3DMM parameters and preserves the identity and details of the original image.
2. The proposed method leverages the strengths of the Diff-AE backbone and ControlNet to synthesize photo-realistic facial images corresponding to the desired pose, expression, and lighting while only trained with 2D in-the-wild images.

**Limitations:**

1. The proposed RSM training strategy has already been proposed in Masked Autoencoders. Despite the authors stating that the purpose is not for representation learning, it cannot be considered a novel contribution. And the Exp-FaceNet is essentially ControlNet using 3DMM parameters as the input conditioning. The contribution of the proposed framework is incremental.
2. The Exp-FaceNet needs to be pre-trained, so the proposed method is not one-shot.
3. The face reconstruction quality in the one-shot explicit editing shown in the supplementary video is unconvincing. The reconstruction results of DiffusionRig seem to preserve identity better and also have more details. For example, in the lighting editing result at 03:23, the result of DiffusionRig is more faithful.

**Suitability:**

2

---

### Official Review · Reviewer_JXex · 2024-05-26

**Rating:** 6
**Confidence:** 4

**Summary:**

In this research, the authors propose a novel diffusion-based editing framework called DisControlFace, which aims to achieve fine-grained control of generative facial image editing. They address the challenge of disentangled conditional control between high-level semantics and explicit parameters by leveraging a Diffusion Autoencoder (Diff-AE) and an Exp-FaceNet. The model is trained using 2D portrait images and demonstrates superior editing accuracy and identity preservation compared to existing methods.

**Strengths:**

1. The framework generates facial images that are visually realistic and faithful to the original input. The Diffusion Autoencoder (Diff-AE) and Exp-FaceNet components ensure that the generated images maintain the desired facial appearances and accurately represent the input semantics.
2. DisControlFace ensures consistent semantic details in the generated images. By leveraging the Diff-AE and Exp-FaceNet, the framework maintains the semantic conditioning capability and reduces semantic information shift during the editing process. This helps in preserving the identity and overall coherence of the generated images.
3. Comprehensive experiments demonstrate that DisControlFace outperforms state-of-the-art methods in terms of editing accuracy and identity preservation. The framework generates realistic facial images with a high level of accuracy, making it a reliable tool for generative facial image editing tasks. By the way, the demo video is awe-inspiring for me.

**Limitations:**

I am very curious about how this framework works in the case of a profile face. The authors are encouraged to discuss this case.

**Suitability:**

3

---

### Meta-Review · Area_Chair_UUWx · 2024-06-27

**Recommendation:** Accept (Poster)
**Confidence:** 4

**Metareview:**

This paper was reviewed by four experts in the field. The paper received mixed reviews BR, BA, WA, A. Reviewers like the generation of realistic facial images and the effective disentangling control achieved by the proposed framework. The only negative reviewer raises cocnerns on missing details and typos, which the AC believes did not challenge the contribution of this paper. Based on the scores, we recommend the acceptance of this paper.

---

### Meta-Review · Senior_Area_Chairs · 2024-07-10

**Recommendation:** Accept (Poster)
**Confidence:** 4

**Metareview:**

This paper received mixed ratings initially. After rebuttal, all the reviewers are satisfied with the rebuttal and 3 of them tend to accept the paper while another increased score form R to BR. SAC and AC carefully check the reviews and rebuttal and recommend acceptance of the paper.